# Local and Systemic Effects of Bioactive Food Ingredients: Is There a Role for Functional Foods to Prime the Gut for Resilience?

**DOI:** 10.3390/foods13050739

**Published:** 2024-02-28

**Authors:** Emma F. Jacquier, Marcel van de Wouw, Elena Nekrasov, Nikhat Contractor, Amira Kassis, Diana Marcu

**Affiliations:** 1Neat Science, 1618 Chatel-Saint-Denis, Switzerland; amira@theneatscience.com; 2Department of Pediatrics, University of Calgary, Calgary, AB T2N 1N4, Canada; 3Amway Innovation and Science, Ada, MI 49355, USA; elena.nekrasov@amway.com (E.N.); nikhat.contractor@amway.com (N.C.); 4School of Molecular Biosciences, College of Medical, Veterinary & Life Sciences, University of Glasgow, Glasgow G12 8QQ, UK; d.marcu.1@research.gla.ac.uk

**Keywords:** microbiota, resilience, functional foods, gut health, hormesis, fibres, phytonutrients, nutrition

## Abstract

Scientific advancements in understanding the impact of bioactive components in foods on the gut microbiota and wider physiology create opportunities for designing targeted functional foods. The selection of bioactive ingredients with potential local or systemic effects holds promise for influencing overall well-being. An abundance of studies demonstrate that gut microbiota show compositional changes that correlate age and disease. However, navigating this field, especially for non-experts, remains challenging, given the abundance of bioactive ingredients with varying levels of scientific substantiation. This narrative review addresses the current knowledge on the potential impact of the gut microbiota on host health, emphasizing gut microbiota resilience. It explores evidence related to the extensive gut health benefits of popular dietary components and bioactive ingredients, such as phytochemicals, fermented greens, fibres, prebiotics, probiotics, and postbiotics. Importantly, this review distinguishes between the potential local and systemic effects of both popular and emerging ingredients. Additionally, it highlights how dietary hormesis promotes gut microbiota resilience, fostering better adaptation to stress—a hallmark of health. By integrating examples of bioactives, this review provides insights to guide the design of evidence-based functional foods aimed at priming the gut for resilience.

## 1. Introduction

The notion of food as medicine can be traced back thousands of years, when the use of plants, herbs, and foods to treat disease and restore health was commonplace. Today, it is well understood that diet interacts with human physiology and health [1,2,3]. Healthy dietary patterns that favour plant-based foods have been shown to offset the risk of chronic disease [2,4]. Furthermore, components of plant-based diets have been shown to influence human physiology via effects on the gut microbiome [5,6,7]. In recent years, the field of research into the effects of various bioactives on gut microbiota and wider physiology has grown rapidly [8]. The innovation opportunity to design functional foods to support a resilient gut microbiota with a view to safeguarding against wider pathologies associated with gut dysbiosis extends the concept of functional foods into a new sphere. A pivotal aspect of this exploration is the recognition of the body’s capacity to adapt to stress through the resilience of the gut microbiota. The gut microbiota, a complex community of microorganisms residing in the gastrointestinal tract, plays a crucial role in modulating local and systemic responses to stressors [9]. Bioactive food ingredients, through their interactions with the gut microbiota, may contribute to microbiota resilience, thus supporting the body’s adaptive capacity.

Educated consumers are becoming more aware of the potential beneficial links between bioactive food ingredients, the gut microbiota, and wider physiology, leading them to seek out functional foods with active ingredients which support gut health [10,11,12,13]. For food innovation specialists and product developers, this represents both an opportunity and a challenge. On the one hand, there is a huge opportunity to combine blends of active ingredients, as a simple and effective intervention for gut health, via incorporation into functional food and beverage carriers. On the other hand, product developers must navigate the myriad of bioactives under investigation and their potential influence on the microbiota and wider physiology, along with varying levels of evidence and efficacy. In addition, the emergence of studies investigating the bidirectional relationship between gut microbiota and broader organ, or systemic, health brings both innovation opportunities and new product development challenges. However, the careful selection of bioactives in the design of functional foods opens avenues for developing strategies to prime the gut microbiota resilience and thus promote broader well-being.

This narrative review is intended to guide food scientists and product developers in their understanding of the role of the gut microbiota on host health, the notion of gut microbiota resilience, and the evidence related to the broad gut health benefits of popular dietary components, functional ingredients, and bioactives. Importantly, this study will explore the potential local and systemic benefits of popular and emerging ingredients. Such deliberate consideration of the evidence may contribute to the improved design of functional foods which may help to prime the gut microbiota resilience, influence broader physiology, and potentially influence overall well-being.

## 2. Hormesis in the Context of Diet: Priming the Gut to Become More Resilient

An adequate response to stress has been defined as one of the main hallmarks of health [14]. Health is continuously threatened by multiple sources of stress. Organisms employ diverse strategies, including homeostatic resilience, hormesis, repair, and, when feasible, the regeneration of impaired tissues and organs, to attain biological stability. The robust resilience of a “healthy” microbiota may further contribute to adequate responses to stress and serve as a safeguard against various pathologies associated with dysbiosis, such as inflammatory bowel disease, metabolic syndrome, cardiovascular dysfunctions, depression, asthma, rheumatoid arthritis, colon cancer, and autism spectrum disorders [15].

Sommer et al. [16] and others have discussed how perturbations during the life course may impact the gut microbiota. This is because the gut microbiota ecosystem has its own homeostasis in both health and disease. As part of that homeostasis, perturbations (or “challenges”) will typically, briefly, alter the gut microbiota, after which it returns to its “normal” homeostasis. This capacity for self-regeneration, or an ability to revert back to a stable state, has been described as microbiota resilience [15,16]. In Figure 1, scenario (a) shows a recovery to “normal” health following a perturbation, whereas (b) shows an inability to recover, resulting in a gut microbiota dysbiosis. Well-designed functional foods, consumed on a frequent basis, may help to prime the gut in order to support gut microbiota recovery as in situation “a”, thus promoting microbiota resilience.

Sommer et al. postulates that scenarios a and b are plausible according to the resilience of the gut microbiota. In scenario a, the microbiota returns to its stable state after a perturbation (e.g., medication, short-term dietary changes, or sickness). In scenario b, the host was unable to “bounce back”, leading to dysbiosis and its associated pathologies (Figure 1). Scenarios a and b may have repercussions in the case of human physiology where the gut immune system and gut–organ axes are in continuous communication with the microbiota.

Thus, stress adaptation and resilience are pivotal components of health, encompassing the ability of an organism to respond and adapt effectively to stressors, ultimately promoting overall well-being. This adaptability is particularly evident in the context of the gut microbiota, which plays a crucial role in immune function, nutrient metabolism, and overall gastrointestinal health [17]. Gut microbiota resilience and adaptation to stress involve complex interactions with the host’s stress response systems, immune system, and gut–organ axes, as well as genetic and environmental factors, enabling the microbiota to maintain or return to homeostasis after disturbances [18]. Nevertheless, the integration of gut microbiota resilience into the framework of health underscores the significance of diet as a key modulator.

How can we use diet to improve the gut microbiota’s response to perturbations/stress and prime the gut to be more resilient? The hormetic responses from dietary interventions may be the key for improving gut resilience. Hormesis is a biphasic dose–response phenomenon, wherein exposure to low levels of stressors induces adaptive responses that confer resistance to subsequent challenges [19]. In the context of diet, hormetic effects may arise from exposure to phytochemicals, dietary polyphenols, and other bioactive compounds present in fruits, vegetables, and certain spices (see Figure 2). Noteworthy studies support the assertion that hormetic stressors derived from certain dietary components (dietary hormetins) can exert a positive influence on gut health [19,20]. Dietary hormetins were shown to stimulate and strengthen adaptation and repair systems in cells and tissues, leading to the maintenance of a healthy status [21]. For instance, polyphenols present in foods like green tea and berries have been shown to induce mild oxidative stress, prompting cellular adaptations that contribute to enhanced resilience of the gastrointestinal epithelium [22].

Furthermore, dietary hormetins improve microbiota resilience by altering gut microbiota composition, stimulating beneficial bacteria growth, maintaining microbial equilibrium, increasing diversity and metabolic functions, and enhancing cellular maintenance and repair, which collectively contribute to better health and reduced disease susceptibility [15,23]. Moreover, an optimal and tailored dietary intake of hormetic phytochemicals, encompassing polyphenols, carotenoids, sulforaphane, and various bioactive compounds, has been acknowledged for its capacity to activate intracellular signalling cascades and modulate gene expression, contributing to gut microbiota resilience [24]. This enhanced resilience of the gut microbiota, mediated through the gut–brain axis, influences the body’s response to stress. Figure 1 highlights the dual potential of this interaction, depicting scenarios where adaptive stress responses or maladaptive stress responses are promoted based on the modulation of the gut microbiota.

Beyond the local effects on the gut, hormetic responses triggered by dietary compounds extend to systemic health. These adaptive mechanisms include the activation of cellular stress response pathways, such as the Nrf2 antioxidant pathway, which plays a pivotal role in maintaining cellular homeostasis and mitigating inflammation [25]. Hormesis is also implicated in promoting longevity and resilience to age-related diseases. By inducing stress resistance at the cellular and molecular levels, hormetic responses may contribute to the prevention or attenuation of chronic diseases associated with ageing, such as neurodegenerative disorders and metabolic syndromes [26]. In conclusion, understanding the role of hormesis in the context of diet provides valuable insights into the mechanisms underlying its positive impact on both gut and systemic health. Harnessing the potential benefits of hormetic responses from dietary components opens avenues for developing strategies to prime the gut resilience and thus promote overall well-being.

Theoretically speaking, dietary patterns, foods, and carefully designed functional foods, may, over time, contribute to priming the body to respond to daily stressors: A future perturbation (or challenge) has a smaller impact on the gut microbiota or host physiology (e.g., the organism is less affected by daily stressors, or sickness is less intense/impactful).The recovery following a perturbation (or challenge) is shorter (i.e., quicker recovery);In cases where recovery after a perturbation (or challenge) is incomplete, the food or supplement helps to resolve some of the long-term deficits (e.g., reducing inflammation).

## 3. The Impact of Gut Microbiome on Host Health

Animals and humans live and have evolved in close association with microbes [27]. The human microbiome comprises collective genomes of microorganisms, such as bacteria, archaea, viruses, and eukaryotic microbes, inhabiting in and on us [28]. The gut microbiota codevelops with the host from birth and lives symbiotically within various habitats of the human body, such as the oral cavity, genital organs, respiratory tract, skin, and gastrointestinal system [29]. The gastrointestinal tract accommodates the densest number of microbes, being collectively called the gut microbiota and predominantly composed of bacteria from *Bacillota*, *Bacteroidota,* and *Actinomycetota* phyla [30]. Early studies demonstrated that the infant gut microbiota shows major compositional shifts in its bacterial communities as it develops and reaches and adult-like composition around 3 years of age [31]. Subsequent shifts in the microbial landscape were associated with life events, such as disease, ageing, dietary changes, or antibiotic use [29,32].

### 3.1. The Functional Relevance of Microbiota

Accumulating evidence suggests that the gut microbiota plays a key role in the initiation and progression of metabolic diseases [33]. A transplantation of gut microbes from conventionally reared mice into germ-free mice resulted in an increased accumulation of adipose tissue in the germ-free mice, despite controlling their food intake, thus demonstrating that gut microbiota transplantation is capable of leading to an increased body fat content in mouse models [34,35]. Moreover, the intestinal microbial composition was shown to be different between obese and lean individuals, with obese persons having a lower prevalence of Bacteroidota but a higher prevalence of Bacillota [36]. Compositional variation in the intestinal microbiota has also been observed when comparing faecal samples from type 2 diabetes (T2D) patients versus healthy individuals [37]. The T2D samples had a lower abundance of bacteria from the genus *Lactobacillus* and higher abundance of *Bifidobacterium*. These studies illustrate the significance of microbiota for potentially offsetting the risk of chronic disease.

Beyond specific diseases, evidence suggests that the gut microbiota significantly contributes to general well-being, especially in later life. Research across various model organisms, as well as in humans, demonstrates the role of the gut microbiome in well-being and longevity. Generally, as organisms age, the diversity of their gut microbiota decreases and health-promoting bacteria decline, while opportunistic bacteria and overall microbial numbers increase [38]. Human studies also indicate age-related compositional changes in the microbiota, suggesting a potential causal role in late-life health [39]. Centenarians were shown to overcome those changes by maintaining gut microbiota diversity and by increasing the prevalence of health-associated groups (e.g., *Akkermansia* and *Bifidobacterium*) [40]. These studies suggest that the gut microbiota shows compositional changes that correlate age and disease, implying that interventions which target the gut microbiota have the potential to improve both health span and lifespan.

### 3.2. What Is a “Healthy” Gut Microbiota? A Holistic Exploration of Dysbiosis-Induced Pathogenesis

Substantial research is dedicated to defining a “healthy” gut microbiota and understanding its connection to host physiological functions. However, the concept of a “healthy” gut microbiota remains complex and evolving within scientific discourse, lacking a universally agreed-upon definition due to variations among individuals. Nevertheless, certain characteristics associated with gut microbiota composition are often considered indicative of a potential state of health. Under healthy conditions, the gut microbiota exhibits stability, resilience, and a symbiotic relationship with the host [41]. A “healthy” gut microbiota is generally characterised by a balanced and diverse microbial community, essential for maintaining gut barrier integrity, nutrient metabolism, and immune system modulation [41].

In a state of homeostasis, the gut microbiota performs essential functions, such as aiding in the digestion of complex carbohydrates, the extraction of nutrients from food, and the biosynthesis of bioactive molecules, including vitamins, amino acids, lipids, or short-chain fatty acids [34]. Moreover, the gut microbiota not only shields the host from external pathogens by producing antimicrobial substances, but also contributes significantly to the development of intestinal mucosa and the immune system [42]. Dysbiosis can instigate disease processes through multiple mechanisms. Disruption of the gut barrier, altered immune responses, and dysregulation of metabolic pathways are among the key contributors [43,44].

But how might gut-confined microbes have such systemic effects on health? The gut microbiota can produce and regulate signals that reach the circulation and modulate whole-organism physiology. These signals can be either microbiota-derived or host-derived. Microbial signals can be structural components of bacteria or metabolites. Structural components such as lipopolysaccharide (LPS), peptidoglycan, and flagellin are recognised by pattern-recognition receptors on epithelial and immune cells and they are generally not diffused across the epithelial barrier [45]. It has been reported that LPS might reach the circulation via co-transport with chylomicrons, which was suggested to have implications for inflammation and obesity [46]. An altered composition of gut bacteria has also been shown to catabolise tryptophan into the metabolite indole-3-aldehyde, which has been associated with inflammatory bowel disease [47]. Moreover, SCFAs, such as butyrate, propionate, and acetate, generated from the saccharolytic fermentation of dietary fibre, can act as signalling molecules at distant body sites. For example, acetate, produced from dietary fructose, is a source for supplying acetyl-CoA, which can then trigger hepatic de novo lipogenesis [48]. Thus, bacterially sourced molecules can systemically affect host health.

As well as containing bacteria that produce signals, the gut is the largest endocrine organ in the human body, containing specialised hormone-producing enteroendocrine cells [49]. In mice, SCFAs have been shown to activate receptors on those cells, which then triggers the secretion of gut peptides such as glucagon-like peptide (GLP-1) and peptide YY (PYY) [50]. GLP-1 is known to regulate pancreatic function, insulin release, and appetite, while PYY has been shown to increase energy harvest from the diet [51,52]. Human studies have demonstrated that the administration of fermentable fibres induces changes in the gut microbiome, which, when modulated by fructooligosaccharides (FOS), is suggested to enhance satiety, reduce hunger, and elevate GLP-1 and PYY levels [53]. Thus, microbes that reside in our gut generate molecules that can influence host endocrine signalling, and this may have consequences for wider systemic health.

## 4. The Interplay between Diet, Microbiota, and Human Health

The gut microbiota plays a pivotal role in maintaining human health, and dysbiosis can contribute to the development of various diseases. Research has established that the human diet significantly shapes the diversity and the relative abundance of microorganisms in the gut microbiota [54]. Macronutrients, micronutrients, and bioactive compounds present in food exert differential effects on the microbial community, contributing to either stability or dysbiosis. The diet is a crucial modifiable factor influencing the gut microbial communities, with over 50% of microbiota variation attributed to dietary changes [55,56]. It is well-established that poor nutritional status, including undernutrition, micronutrient inadequacies, and excessive calorie intake, has been associated with several negative health outcomes such as cancer, cardiovascular disease, type II diabetes, digestive discomfort, and poor mental health outcomes (e.g., anxiety, depression) [57].

Dietary patterns that adhere to established guidelines for a healthy diet are associated with a reduced risk of chronic disease, a healthier body weight, and a healthier gut microbiome profile. For example, the Mediterranean diet, characterised by a high consumption of fruits, vegetables, whole grains, and olive oil, promotes the growth of beneficial bacteria and reduces the abundance of potentially pathogenic microbes [58,59,60]. This emphasizes the influence of holistic dietary patterns on the gut microbiome. On the other hand, Western diets, characterized by a high intake of processed foods, sugars, and saturated fats and a low fibre content, have been extensively linked to an increased risk of chronic diseases, including cardiovascular disease, obesity, T2D, and certain cancers [61,62,63], as well as the development of gut dysbiosis [64]. Furthermore, processed foods and animal-derived foods have been associated with endotoxin production, whereas a plant- and fish-based diet has been linked to short-chain fatty acid synthesis and optimal nutrient metabolism [65]. Such studies also report that the same diet, based on animal-derived and processed foods, was associated with increased intestinal inflammation, poorer gut health, and alterations in the gut microbiota [65].

A more diverse plant-forward diet, emphasizing plant foods without eliminating animal foods, will provide a wider array of different nutrients for the gut microbiota, therefore potentially allowing a wider range of different gut microbes to inhabit the gut microbiota. This was exemplified by initiatives such as the American Gut Project, which found associations between an increased variety of plant species (participants consuming > 30 different plant species per week) and microbiota diversity versus those participants consuming < 10 different plant species per week [60]. In tandem, a more diverse diet is also likely associated with a more stable gut microbiota [66], a hallmark of gut microbiota resilience [16]. The gut microbial ecosystem is relatively resilient to individual short-term changes, which include diet and dietary supplements. For instance, even though short-term dramatic dietary interventions may change the composition of the gut microbiota, these changes are transient and dissipate within a few days [67]. Importantly, long-term dietary patterns are more linked to a person’s unique gut microbiota composition [68]. This suggests that long-term, sustainable dietary interventions are necessary to incur changes in the gut microbial ecosystem that may potentially confer health benefits. These sustainable interventions, featuring carefully selected foods, could be thought of as priming the gut for resilience [69].

Improving dietary diversity for some sectors of the population is complex, since consuming a diverse diet can be economically costly [70]. Energy-dense foods provide energy to the diet at a more affordable cost than nutrient-dense foods such as lean meats, dairy products, and fruits and vegetables [71]. Knowledge about healthy foods and meal preparation can be lacking, which is also associated with poorer-quality diets [72]. In addition, fresh plant-based foods can be difficult to access and consume “on-the-go”. This is where functional foods can deliver targeted nutrition in a convenient way. Functional foods refer to food products that not only provide basic nutrition but also contain bioactive components that offer additional health benefits beyond their nutritional content [73]. Functional ingredients may have physiological benefits or may contribute to reducing the risk of chronic diseases when consumed as part of a regular diet [74,75]. It is theoretically possible, therefore, to carefully design functional foods that consist of a variety of different bioactive components, which could support a healthy gut microbiota and potentially improve gut and systemic health, if consumed on a regular basis.

## 5. A Multi-Pronged Approach to the Design of Functional Foods for Gut and Systemic Health

How, then, might functional food developers set about selecting ingredients with the highest probability of influencing the functional parameters of gut and systemic health and shaping the microbiota towards a healthy state? In recent history, the field of nutrition science evolved from focusing on how intakes of individual nutrients impact the risk of disease, to examining dietary patterns and their influence on health. Importantly, there is now an increasingly larger emphasis on a “holistic approach” in terms of diet because (1) many of the nutrients that we eat have synergistic effects with each other on health, such as vitamin C (ascorbic acid) enhancing the absorption of iron [76], and (2) many of the foods that we eat naturally consist of a complex matrix of various nutrients [57]. Global dietary guidelines today also tend to emphasise the notion of varied and diverse diets, favouring fruits, vegetables, and wholegrains, the consumption of which must be maintained over the life course.

Despite nutritional strategies to modulate the gut microbiota towards a resilient state being an emerging field of study, it is theoretically possible to design food-based interventions that may act on functional parameters of gut health along with shaping the gut microbiota towards a resilient, healthy state [23]. Although product developers may be well-equipped to select ingredients that pose the required technical properties for incorporation into their product matrix, choosing the right bioactives that impact the gut microbiota and beyond can be challenging. Despite the growing body of evidence supporting the relationship of specific functional ingredients with gut health, the field itself is a complex one, as the gut holds a bidirectional relationship with other organs and systems in the body. Consequently, different bioactive ingredients may act directly or indirectly on different gut outcomes, having local effects or broader systemic effects. Developing an understanding of how certain bioactive components may act locally, or systemically, would help product developers to innovate in the gut health field. This could be particularly useful for innovation purposes when new concepts are developed for testing with consumers.

## 6. Ingredients for Functional Foods and Beverages with Promising Gut-Health-Related Outcomes

This section discusses a range of potential ingredients with gut health benefits, ranging from well-studied ingredients through to emerging ingredients with increasing levels of scientific evidence. In particular, this section distinguishes between the local, systemic, and potential hormetic effects of functional ingredients.

### 6.1. Dietary Fibres

Dietary fibres, a diverse group of non-digestible carbohydrates derived from plant cell walls, represent an integral component of human nutrition with profound implications for health [77]. The latest consensus affirms that dietary fibre consists of carbohydrate polymers that undergo neither digestion nor absorption in the human intestine [78]. Instead, they proceed to the large bowel, where the colonic microbiota partially or fully ferments them. Dietary fibre variants can be conventionally categorized based on their solubility. Early studies indicated that the solubility of fibres is key to its effects on gastrointestinal health, where insoluble fibres were thought to reduce transit time and decrease the absorption of some nutrients such as glucose [79,80], and nonetheless, the concept of fibre solubility has proved controversial in recent times. This is due to (1) different methods for measuring fibre solubility giving inconsistent results, (2) differences in gastrointestinal pH between individuals impacting fibre solubility, (3) solubility alone not being the sole predictor of the physiological effects of fibre, and (4) the fact that dietary fibres are often a mix of individual soluble and insoluble fibres [81].

Nonetheless, fibres, both soluble and insoluble, contribute to local gut health and systemic overall health through various mechanisms [82,83,84]. It is important to note that some reviews have already hypothesized that dietary fibres may help to prime the gut for increased gut microbiome resilience, and that the incorporation of dietary fibres into daily nutrition is crucial for maintaining a balanced gut microbiota, optimizing intestinal physiology, and mitigating the risk of various systemic diseases [15].

#### 6.1.1. The Local Gut Health Effects of Dietary Fibres

Dietary fibres emerge as essential components for the promotion and maintenance of optimal gut health. They are able to significantly modulate the local gut environment by influencing the gut microbiota composition, enhancing mucosal integrity, and modulating immune responses [85,86]. Insoluble fibres, like cellulose, add bulk to the stool, aiding in regular bowel movements and preventing constipation [87]. Similarly, fibre powders, particularly those containing partially hydrolysed guar gum (PHGG), offer a range of health benefits. Being a soluble fibre, PHGG absorbs water in the intestines, forming a gel-like substance [88]. This adds bulk to the stool, promoting peristalsis and facilitating regular bowel movements. This effect is particularly beneficial for individuals struggling with constipation. Moreover, two randomised clinical trials showed that its prebiotic properties, along with its ability to regulate bowel movements and reduce bloating, make PHGG a promising dietary intervention for individuals with irritable bowel syndrome (IBS) [89,90].

Additionally, soluble fibres can act as “prebiotics”, enhancing gut microbiota resilience by serving as substrates for beneficial bacteria in the gut and contributing to a balanced and diverse gut microbiota. A consensus definition recognises prebiotics as fermentable dietary compounds that, upon consumption, specifically stimulate the growth and/or activity of beneficial gut bacteria [91]. Prebiotics, such as fructans (FOS and inulin) and galactans (galactooligosaccharides, or GOS), are fermented by bacteria like *Bifidobacteria* and *Lactobacilli*, promoting their growth [91]. This fermentation process produces short-chain fatty acids (SCFAs), particularly butyrate, known for its role in maintaining gut health by supporting the integrity of the gut epithelium, enhancing the mucosal barrier function, and reducing inflammation [92]. This contributes to overall gut health and may help to prevent conditions such as inflammatory bowel diseases [92]. SCFAs alter the intestinal environment by reducing the pH, thereby preventing the overgrowth of pH-sensitive pathogenic bacteria [93,94]. Additionally, they mitigate protease activity, linked to the generation of harmful metabolites like ammonia, a potentially carcinogenic byproduct of protein fermentation. SCFAs actively participate in the intestinal defence system against pathogens and toxic compounds [95]. SCFAs contribute to enhanced gut barrier function by regulating the expression of mucins, which constitute primary physical barriers against pathogens [85,96] and improve gut permeability [97,98].

Furthermore, algal-based polysaccharides, such as fucoidan, are gaining attention as potential prebiotics due to emerging evidence of their potentially beneficial effects, including immunomodulation and antioxidative, antiviral, and antimicrobial properties, as well as improving serum dyslipidaemia and gut health in a rat model [99,100,101].

#### 6.1.2. The Systemic Health Effects of Dietary Fibres

From metabolic benefits to cardiovascular protection, weight management, and immune modulation, incorporating dietary fibres into one’s diet emerges as a fundamental strategy for promoting and maintaining overall health [82]. SCFAs produced during fibre fermentation exert anti-inflammatory effects, influencing immune responses and reducing systemic inflammation [102].

These SCFAs are not only associated with gastrointestinal health, but also with improved metabolic health and reduced systemic inflammation [103]. There is strong evidence showing that SCFAs can impact other organ systems, such as the immune system, host energy metabolism, and the brain [104,105,106]. Moreover, SCFAs elicit the secretion of satiety hormones, GLP-1 and PYY [107]. Both hormones influence the hypothalamus to promote satiety. Moreover, SCFAs can undergo conversion into glucose through intestinal gluconeogenesis (IGN). This process activates adipocytes to produce leptin, contributing to enhanced satiety and providing a preventative effect against obesity [108]. Furthermore, increased IGN by SCFAs inhibits hepatic gluconeogenesis, leading to improved glucose tolerance. Furthermore, systematic reviews and meta-analyses of case–control and cohort studies demonstrated that non-digestible carbohydrates from plant sources decrease both the development and progression of colorectal cancer [109]. Additionally, a randomised double-blind controlled trial investigated the long-term effects of resistant starch supplementation in patients with an elevated risk of hereditary cancers (Lynch Syndrome) [110]. SCFAs also exhibit immunomodulatory effects by regulating antimicrobial peptide (AMP) synthesis, expanding regulatory T cells, and influencing myeloid cell function, thereby contributing to reduced inflammation. Consequently, the cumulative impact of non-digestible carbohydrate (NDC)-induced SCFA production is associated with important markers related to metabolic diseases, including obesity, cardiovascular diseases, and diabetes.

Although there are many studies showing that most (if not all) fibres increase SCFA levels, it is still unclear which fibres are most effective at increasing SCFA levels [111,112,113,114,115,116]. This connection between gut health and systemic well-being clearly highlights the role of dietary fibres in preventing chronic inflammatory conditions. In summary, dietary fibres, through their soluble or insoluble characteristics and the fact that they are fermentable or poorly fermentable, provide diverse characteristics for ingredients that may contribute to gut health. In addition, the selection of ingredients which may promote the generation of SCFAs may contribute significantly to priming the microbiota for resilience, gut health, and systemic well-being, and may thus be considered a key ingredient choice to potentially promote overall health when designing functional foods.

### 6.2. Local and Systemic Effects of Probiotics and Postbiotics

Over the last twenty years, there has been a global surge in the availability of probiotic-containing foods, beverages, and supplements [117]. Dairy remains the largest sector of the market to incorporate probiotic bacteria into a variety of dairy matrices [118]. Probiotics are defined as “live microorganisms that, when administered in adequate amounts, confer health benefits to the host” [91]. The most common probiotics include strains of bacteria, such as *Lactobacillus* and *Bifidobacterium*, and yeast like *S. boulardii* [91]. The proposed mechanisms underlying the effects of health benefits of probiotics involve alterations in the composition and function of the intestinal microbiome. Probiotics are known to generate antimicrobial agents or metabolic compounds, exerting suppressive effects on the growth of other microorganisms [119,120]. Additionally, they engage in competition for receptors and binding sites on the intestinal mucosa with other microbes present in the intestinal environment, as highlighted by Collado et al. in 2007 [121]. Probiotics exhibit the capacity to modulate intestinal immunity, influencing the responsiveness of both intestinal epithelial cells and immune cells to microbes within the intestinal lumen. Numerous studies have demonstrated the ability of *L. reuteri* to regulate cytokine production by human immune cells [122].

The integrity of the intestinal barrier is enhanced by *Lactobacillus* strains, potentially contributing to the maintenance of immune tolerance and a reduction in the translocation of bacteria across the intestinal mucosa. This, in turn, may impact disease phenotypes such as gastrointestinal infections, irritable bowel syndrome (IBS), and inflammatory bowel disease (IBD) [123,124,125]. A clinical study observed reduced pain and flatulence in irritable bowel syndrome (IBS) patients receiving a 4-week treatment with a rosehip drink containing *L. plantarum* DSM 9843 [126]. The presence of *L. plantarum* in rectal biopsies of patients correlated with improved clinical symptoms, accompanied by decreased enterococci levels in faecal specimens. Another investigation focused on patients with diarrhoea-dominant IBS (IBS-D) who experienced symptomatic relief with a probiotic mixture (*L. acidophilus*, *L. plantarum*, *L. rhamnosus*, *Bifidobacterium breve*, *B. lactis*, *B. longum*, and *Streptococcus thermophilus*) [127].

In addition to their local beneficial effects on intestinal integrity, immunity, and health, probiotics and their respective postbiotics have been shown to impact systemic immunity and health through a complex communication system between the gut and the organs. For instance, the microbiota–gut–brain axis (MGBA) is constantly relaying and deciphering information between the periphery and the brain [128]. To maintain homeostasis, the CNS constantly responds to environmental signals conveyed through the vagus nerve, a pivotal participant in the communication within the MGBA [128]. Peripheral cytokine production triggers the vagal anti-inflammatory reflex, leading to the production of acetylcholine, which subsequently prevents tissue damage by excessive cytokine release [129]. Recent research pointed out alterations in gut microbiota [130,131,132,133,134] as well as vagal tone in depressed individuals [135], as well as patients with anxiety disorders [136] and schizophrenia [137]. Some probiotics, such as *Bifidobacterium,* were shown to signal to the brain via vagal pathways [138,139]. Accordingly, when the vagal nerve is cut, the effects of probiotics on the brain and behaviour are blocked [140,141,142].

Heat-killed bacteria have emerged as an alternative to conventional probiotics. Generally, compounds containing inanimate microorganisms and/or their components, which, when administered in sufficient amounts, confer benefits to consumers, are termed postbiotics [143]. The rise of postbiotics emerged as a resolution to address the limitations associated with probiotics [144]. These limitations include factors such as unclear molecular mechanisms, short-lived and niche-specific actions, the potential for developing antibiotic resistance or the transfer of virulence genes, ambiguous beneficial effects, challenges related to maintaining viability and stability in the production process, obstacles for the colonisation of commensal gut microflora, and the ability to induce opportunistic infections and inflammatory responses [145,146]. Moreover, the integration of probiotics into complex multi-component functional foods poses practical limitations primarily related to formulation challenges and stability issues. Achieving the desired concentration of viable probiotics throughout the shelf life of the product can be challenging, as factors like food matrix interactions, processing conditions, and exposure to environmental elements may affect probiotic viability and functionality [147]. It is noteworthy that non-viable microorganisms were shown to maintain their potentiality to exert beneficial effects on the host at the intestinal level in vivo, contributing to the development of safer preparations with improved pharmaceutical properties.

#### 6.2.1. Heat-Inactivated Lactobacillus

A study by Saito et al. [148] demonstrated that oral administration of heat-inactivated (HI) preparations of *L. brevis SBC8803* in rats upregulates the acyl ghrelin concentration, which in turn increased the ratio of acyl to des-acyl (inactive) ghrelin in the blood. Apart from this, supplementation with HI *Lactobacillus casei* subsp. casei 327 was shown to promote serotonin (5-HT) synthesis in colonic mucosa, thus improving GI motility in mice [149]. Heat-inactivated *L. reuteri* 100-23 was shown to stimulate an immunoregulatory response in mice, which suppresses immune responses directed against it in the bowel, thus enabling the bacteria to persist in the gut and maintain a commensalistic relationship with the host [150]. HI *L. planetarium* L-137 supplementation in fish modulates their intestinal morphology (i.e., increased villus height, villus surface area, and muscularis mucosae) and increases the relative *Bacteroidota*, *Delftia*, *Brevinema,* and *Elizabethkingia* abundances in the gut microbiota [151,152]. In humans, one uncontrolled study supplementing HI *L. planetarium* L-137 for 12 months showed an increased relative abundance of gut microbial Bacteroidota and a reduced abundance of Bacillota and *Clostridium* sub-cluster XIVa compared to the baseline [153].

On a systemic level, HI *L. planetarium* L-137 supplementation increases interferon (IFN) ß gene expression in the spleens of pigs [154] and, reduces the gene expression of inflammatory markers in white adipose tissue in mice receiving a high-fat diet [155], as well as in adipose tissue in a rat model of metabolic syndrome [156]. Daily administration of HK *L. planetarium* L-137 in human subjects increases self-reported quality-of-life scores and affectsTh1-related immunity [157], increases IFNß levels [158], and reduces upper respiratory tract infection severity and duration [159].

#### 6.2.2. Heat-Inactivated Bifidobacterium

A previous study demonstrated that the HI probiotic *Bifidobacterium animalis* subsp. *lactis* CECT 8145 (Ba8145) decreased the fat content by more than 30% in the model organism *Caenorhabditis elegans*, supporting the idea that nonviable cells retain probiotic efficacy [160]. Moreover, another study shows that HI Ba8145 intake ameliorates mesenteric adiposity and dyslipidaemia, increases energy expenditure and lean mass, and improves insulin sensitivity in obese rats with metabolic syndrome [161]. Finally, a comparison of live and HI cells of *B. breve* M-16V administered to gnotobiotic mice showed immune-modulating effects that suppressed pro-inflammatory cytokine production in spleen cells and affected intestinal metabolism for both live and heat-killed treatments; however, the live cells exhibited a stronger effect [162]. In humans, in a double-blind, placebo-controlled trial, patients with IBS were recruited from 20 study sites in Germany and randomly assigned to receive either capsules containing HI *B. bifidum* HI-MIMBb75 cells or a placebo for 8 weeks [163]. Their results indicated that just over 30% of subjects in the *B. bifidum* HI-MIMBb75 met the primary endpoint of improvement of abdominal pain and adequate relief of overall IBS symptoms being fulfilled in at least 4 of 8 weeks during treatment, versus 22% in the placebo group. This study demonstrated, in a real-world setting, that specific beneficial bacterial effects are mediated independently of cell viability.

#### 6.2.3. Heat-Inactivated *Saccharomyces cerevisiae* Fermentate

Yeasts have been extensively used as probiotics due to their nutritional value and the multitude of bioactive compounds that they contain. *S. cerevisiae* is the most investigated eukaryotic microorganism [164] because this species has been widely used in the food and alcoholic beverages industry, as well as in the pharmaceutical industry [165]. EpiCor^®^ is an emerging functional ingredient derived from a heat-inactivated dried fermentate produced by a specialised fermentation of *S. cerevisiae*. EpiCor^®^ has been found to influence gut health by promoting the growth of beneficial bacteria and maintaining a balanced gut microbiota [166]. This ingredient has gained attention for its diverse health benefits resulting from a highly complex profile of bioactive compounds, such as yeast metabolites, vitamins, polyphenols, sterols, and phospholipids [167]. EpiCor^®^ has been associated with immune system modulation, demonstrated by an enhancement of innate and adaptive immune responses. Previous in vitro studies in human cells demonstrate that EpiCor^®^ stimulates the activity of natural killer (NK) cells, which play a crucial role in identifying and eliminating infected or aberrant cells [168].

Moreover, this heat-inactivated fermentate of *S. cerevisiae* has been studied in five randomized, double-blind, placebo-controlled clinical trials with over 300 adult participants [169,170,171,172,173]. The findings from these published trials suggest multiple mechanisms of action compared to conventional and nonconventional products. EpiCor^®^ has shown the capacity to positively influence immune responses without inducing excessive suppression or stimulation of overall immune activity. Notably, EpiCor^®^ has exhibited substantial clinical advantages by decreasing the occurrence and duration of symptoms resembling colds and the flu during autumn and winter, irrespective of an individual’s influenza vaccination status [171,173]. The daily consumption of dried yeast ferment significantly increased salivary IgA levels, while at the same time reducing pollen-associated allergy symptoms such as nasal congestion during spring and summer months [172]. EpiCor^®^ was also shown to reduce the secretion of pro-inflammatory cytokines while enhancing the production of anti-inflammatory cytokines, increasing erythrocyte haematocrit levels, and boosting mucosal immune protection [169]. Overall, this dried yeast fermentate product shows potential as a nutritional aid for supporting the immune system in healthy individuals facing acute stress and may also be beneficial for patients with certain chronic conditions that impact both innate and adaptive immune defences.

This was shown to contribute to improved GI symptoms and stool parameters in individuals with symptoms of gastrointestinal discomfort and reduced bowel movements. The improvement of these symptoms was nicely correlated with an improved quality of life and reduced stress levels [166]. Another advantage of EpiCor^®^ administration to support gut health is the relatively low dose needed (500 mg/day), particularly when compared to the high recommended doses for prebiotic fibres [166]. In summary, the heat-inactivated yeast fermentate, derived from *S. cerevisiae*, appears to offer diverse and promising health benefits, including immune system modulation, anti-inflammatory effects, gut health improvement, antioxidant activity, and potential support for respiratory health.

More human clinical trials are needed to further substantiate the benefits of heat-inactivated yeast fermentates, derived from *S. cerevisiae*, and postbiotics in general, as gut primers, along with the further exploration required concerning their effects on local and systemic immunity and health. Nonetheless, they represent a useful option for product developers due to their potential technical advantages and ease of incorporation into various different product matrices. 

### 6.3. Plant-Based Fermented Foods

Fermented foods are created through the microbial activity and enzymatic transformation of food constituents [174]. Throughout the fermentation process, microorganisms such as yeast, fungi, or bacteria break down carbohydrates, such as sugars and starch, into more easily metabolised compounds. This enhances the digestibility and absorption of vitamins, minerals, and other nutrients, turning these products into functional foods associated with potential gut health benefits. Fermented plant ingredients have gained attention due to their potential health benefits, attributed to the transformative action of microorganisms on these substrates. Fermentation can increase the bioavailability of nutrients in plant foods. For instance, the fermentation of grains and legumes has been shown to reduce antinutritional factors and enhance the absorption of minerals, such as iron and zinc [175,176]. Moreover, fermented plant foods often harbour probiotic microorganisms which contribute to gut health by modulating the microbiota, improving digestion, and enhancing immune function [177]. Fermentation breaks down complex carbohydrates and proteins into simpler forms, thus improving the digestibility of plant ingredients. This process may be particularly beneficial for individuals with sensitivities to certain components in raw plants [177]. The fermentation of plants also leads to the synthesis of bioactive compounds. For example, the fermentation of tea leaves produces polyphenols and catechins, which are known to induce a multitude of health benefits by priming the body to become more resilient to stress through hormesis [178]. The potential use of fermented foods as source of psychobiotics is gaining popularity among consumers for their possible therapeutic function on the brain [179]. For example, black carrots fermented with *Lactobacillus plantarum* or *Aspergillus oryzae* prevent cognitive dysfunction by improving hippocampal insulin signalling in amyloid-β-infused rats [180]. Moreover, *Saccharina japonica* algae fermented with *Lactobacillus brevis* BJ20 had memory-enhancing effects via the regulation of the SOD antioxidant system in a randomized, double-blind clinical trial [181]. In conclusion, scientific evidence indicates that incorporating fermented plant ingredients into the diet, or into functional foods, may offer a spectrum of health benefits. These advantages span improved nutrient bioavailability, probiotic effects, the production of bioactive compounds, enhanced digestibility, immunomodulation, and potential mental health benefits.

Fermented grass products, such as grains and oats, have attracted interest for their potential health benefits, particularly in the context of gut health [182]. The production of fermented oat beverages dates back over three decades, aligning with the escalating demand for functional foods. Fermented oat products have been implicated in promoting digestive health through the modulation of gut motility. In vitro studies suggest that fermentation by beneficial microorganisms may enhance the bioavailability of nutrients and contribute to improved digestion [177]. Moreover, β-glucan is commonly used as a functional ingredient in foods as it is readily available as a byproduct of oat and barley. Previous studies demonstrated that β-glucan derived from oats or barley positively influenced gastrointestinal transit time, indicating benefits for gut motility [183]. Moreover, animal studies revealed that β-glucan, specifically extracted from oats, increases satiety-related hormones, such as PYY and NPY, and reduces energy intake and body weight [184,185]. Such findings support the concept of these dietary compounds as potential regulators of the microbiota–gut–brain axis. They are clearly appealing ingredients to introduce into functional foods, or new food concepts for gut health, due to widespread acceptability along with emerging broader systemic health benefits which require further research.

Two rat studies highlight the anti-diabetic effects of fermented oat milk, improving glucose control, lipid metabolism, and liver health and influencing intestinal microbiota composition [186,187]. Notably, one study underscores the synergistic impact of fermented oats and Sidr honey in diabetic rats, with enhanced anti-diabetic effects compared to fermented oats alone [186]. These findings underscore the importance of a holistic dietary approach. Human studies further emphasize the relevance of fermented oats in inducing anti-obesity and anti-cholesterol effects on lipid metabolism [188,189].

Fermented oats are also rich in polyphenols [182]. A striking study by Zhang and colleagues indicates ethyl acetate subfractions (EASs) obtained from fermented oats, prepared with Rhizopus oryzae and exhibiting high polyphenol content, have potential anti-cancer properties [190]. They tested this in vitro, indicating that the EASs inhibited human cancer cell growth without showing toxicity to normal cells, as well as in vivo using mouse models. They also demonstrate that EAS administration leads to increased intracellular reactive oxygen species (ROS) levels and activates the ROS/JNK signalling pathway, contributing to the induction of apoptosis. Overall, these potential anti-cancer benefits may be linked to hormesis, suggesting that priming the gut through dietary-induced hormetic responses may lead to a better adaptation to stress. However, more research is needed to confirm these findings.

### 6.4. Polyphenols and Phytonutrients

#### 6.4.1. Polyphenols, Hormesis Activation, and Adaptation to Stress

During evolution, plants developed biosynthetic pathways for chemicals that prevent microorganisms and insects from eating them. Indeed, more than 100 such biopesticides involved in plant defence have been identified [191]. Such phytochemicals are typically concentrated in exposed vulnerable regions of the plant such as the skin of fruits and the growing buds. While these phytochemicals can exhibit toxicity to mammalian cells at high concentrations, subtoxic doses have been observed to induce adaptive stress responses. Studies across various cell types have proposed hormetic mechanisms of action for specific phytochemicals, emphasising their biological significance as cytoprotectants [20].

These polyphenols induce redox homeostasis, activating vitagene signalling pathways, including Hsp70, HO-1, thioredoxin/thioredoxin reductase, and the sirtuins system [192]. The transcriptional modulation of these cytoprotective genes is mediated by Nrf2, a transcription effector for over 500 cytoprotective genes. Nrf2 activity induces a mild stress response, promoting a healthy physiological state and extending lifespan in pre-clinical models [20]. Chronic Nrf2 stimulation, however, may lead to pathophysiological events, categorizing Nrf2 signalling as a hormetic-like pathway [20]. Increasing evidence shows that plant polyphenols activate Nrf2 signalling, supporting redox homeostasis under stress conditions and suggesting their beneficial properties through adaptive stress responses [22]. As per Naviaux and colleagues [193], the hormetic response occurs when hazards exceed the cellular homeostasis capacity, inducing disruptive molecular changes. An increasing body of evidence shows that most chronic diseases arise from the biological response to a stress factor, not from the initial injury or from the agent of the injury itself. Thus, hormesis strengthens the initial adaptive stress response, eventually mediating protection from a variety of potential injuries (i.e., diseases) and thus priming the body to become more resilient. Recent findings indicate biphasic dose responses of a hormetic type as common effects of plant polyphenols. These polyphenols are considered a “preventive treatment of disease”, inducing biological effects with therapeutic applications through the activation of adaptive responses [194]. Growing evidence suggests that plant polyphenols exert positive effects by acting in a hormetic-like manner, activating adaptive stress response pathways and making the hormesis concept fully applicable to nutrition [195]. This opens innovative new avenues in the future for functional foods.

#### 6.4.2. Reciprocal Interaction between Polyphenols and Gut Microbiota

Studies have shown that plant polyphenols contribute to increased microbial diversity in the gut [196]. Polyphenol-rich diets have been associated with a more abundant and diverse array of bacterial species, fostering a balanced and resilient gut microbiota [196,197]. Plant polyphenols selectively promote the growth of beneficial bacteria, such as *Bifidobacterium* and *Lactobacillus*, while inhibiting the proliferation of harmful pathogens [198,199]. This prebiotic-like effect contributes to the maintenance of a healthy gut microbial community. Furthermore, the gut microbiota actively participates in the metabolism of complex polyphenolic compounds [200]. Microbial degradation leads to the formation of bioactive metabolites, such as phenolic acids and urolithins, which may exert additional health benefits beyond the parent compounds [197]. Thus, the bioavailability and impact of polyphenols are significantly shaped by their microbiota-induced alterations [200]. Numerous studies have been conducted to elucidate the transformation of specific types of polyphenols by gut microorganisms and to pinpoint the responsible microorganisms. It was shown that Clostridium and Eubacterium, from the Bacillota phylum, were shown to be the main genera involved in the metabolism of polyphenols [200].

In conclusion, the reciprocal interaction between polyphenols and the gut microbiota represents a dynamic interplay that holds significant implications for human health. The transformation of dietary polyphenols by commensal microbes not only influences the bioavailability of these compounds but also shapes the composition and functionality of the gut microbial community through exerting prebiotic-like effects. Future research will guide the selection and inclusion of specific polyphenols in functional foods that might contribute to gut health.

#### 6.4.3. Foods Rich in Phytonutrients and Their Associated Gut and Systemic Health Benefits

Isothiocyanates, found abundantly in cruciferous vegetables such as broccoli, watercress, Brussels sprouts, cabbage, Japanese radish, and cauliflower, have been shown to induce the expression of cytoprotective Nrf-2 proteins in liver, intestinal, and stomach cells [201]. Resveratrol, a chemical present in the skin of red grapes and wine, exhibits the capacity to activate stress response pathways and protect cells in rodent models of myocardial infarction and stroke [202]. Polyphenols, such as catechin and theobromine from cocoa and chocolate, were shown to improve arterial stiffness and blood pressure [203,204,205,206]. Green tea is renowned for its high polyphenol content, particularly catechins, which exhibit antioxidant, anti-inflammatory, and cardioprotective effects. Higdon and Frei [207] highlight green tea’s potential in reducing the risk of cardiovascular diseases and certain cancers. Berries are rich in anthocyanins, flavonoids, and other polyphenols, conferring anti-inflammatory and neuroprotective benefits. Previous research suggests that berry consumption may enhance cognitive function and mitigate age-related cognitive decline [208]. Another polyphenol, quercetin, found in apples, berries, citrus fruit, and onions, is known for its antioxidant and anti-inflammatory properties [209]. Quercetin acts as an agent to lower coagulation, hyperglycaemia, inflammation, and hypertension. Multiple clinical studies show that supplementation of quercetin is used to prevent the risk of cardiovascular disease [210]. Citrus fruits, including oranges, grapefruits, lemons, and limes, are renowned for their rich content of bioactive compounds, particularly bioflavonoids (such as hesperidin, naringin, and quercetin) [211]. These compounds contribute not only to the vibrant colours of citrus fruits, but they also play a positive role in cardiovascular health by modulating lipid metabolism, improving endothelial function, and reducing inflammation [212,213].

Moreover, phlorotannins, found in brown seaweeds, have emerged as promising contributors to gut microbiota modulation with potential implications for overall gastrointestinal health. Pre-clinical studies suggest that phlorotannins have beneficial effects on gut microbiota, including improving structure, increasing SCFA levels, protecting intestinal cells, reducing obesity-related alterations, enhancing commensal bacteria growth, and potentially aiding in diabetes management [214,215,216,217]. Carotenoids, such as fucoxanthin and astaxanthin, found in various fruit and vegetables, may also help to improve gut health by modulating the gut microbiota and enhancing immune function, metabolic health, and gut barrier integrity, and may contribute antidepressant and neuroprotective effects through their antioxidant and anti-inflammatory properties [218,219,220,221]. More research is required, however, to better understand the potential properties of phlorotannins and carotenoids in a gut health context.

In recent years, there has been an increasing focus on exploring the biological activities of beets, particularly red beets, and their potential positive impacts on GI health [222,223]. Beet consumption has been considered as an adjunctive therapeutic approach for various pathologies linked to oxidative stress and inflammation. These potential properties and applications are primarily attributed to the presence of a special type of phytochemical found in beets, named betalains, which have demonstrated antioxidant, anti-inflammatory, and chemo-preventive activities in both in vitro and in vivo experiments [223,224]. There are also studies which suggest that these phytochemicals may influence the composition and activity of the gut microbiota. A systematic review conducted by de Oliveira et al. (2021) [225] discusses the relevance of the bioactive compounds found in beets on the microbiome and GI health, suggesting that they act as prebiotics, promoting the growth and activity of beneficial gut bacteria.

Overall, some of the most investigated food sources of phytonutrients are from grapes, green tea, cranberries, blueberries, orange juice, sorghum bran, fungi, coffee, and turmeric [226]. In addition, procyanidins, sinapine, resveratrol, quercetin, catechin, epicatechin, and epigallocatechin-3-gallate are well-investigated phytochemicals [226]. One of the main ways through which polyphenols impact positive health outcomes is by reducing oxidative stress [227] and reducing inflammation [228,229]. In addition, emerging evidence shows that specific dietary polyphenols are metabolized by the gut microbiota, and these by-products are hypothesized to impact physiology and health positively [197,227,230,231]; the link between phytonutrients, gut microbiota, and hormesis provides a fascinating perspective on how these plant-derived compounds contribute to adaptive stress responses and may prime the host physiology to become more resilient.

### 6.5. Spices and Aromatic Herbs

Since ancient times, spices and aromatic herbs have served various purposes, such as preservation, colouring, and enhancing flavours. Traditionally employed in numerous cultures for medicinal purposes, spices have garnered attention from the chemical, pharmaceutical, and food sectors due to their beneficial physiological effects and possible preventative applications in a variety of pathologies [232]. With the advent of highly prevalent chronic diseases and the need for safe, low-cost interventions in healthy ageing, certain herbs and spices are becoming promising candidates for inclusion in functional foods. Spices, rich in fibre, essential oils (EOs), minerals, and pigments, also contain bioactive compounds like phenolic acids, flavonoids andsterols.. EOs in spices, including terpenes, monoterpenes, and sesquiterpenes, also may contribute to their functional properties [233]. However, the functional attributes of spices are largely linked to the presence, type, and concentration of phytochemicals.

As discussed previously, numerous studies have highlighted the functional characteristics of phenolic compounds, particularly flavonoids, including antioxidant [234], antibacterial [232,235], antiviral [236], and anti-inflammatory [237] capacities, as well as their cardioprotective [238] and anticarcinogenic effects [239]. Terpenes are the fundamental component of essential oils. Despite their very different chemical structures, they are all extremely volatile and have been demonstrated to possess multiple functional properties, including antioxidant [240], antimicrobial [233], and antiviral [241] capacities. Therefore, the consumption of herbs and spices demonstrates the potential to locally and systemically modulate host health, primarily, but not only, due to their high phytochemical content.

#### Local Influence of Herbs and Spices on Intestinal and Microbiota Health

The incorporation of spices into foods has a positive impact on gut health. They enhance the secretion of saliva and gastric juices [242] and elevate the concentration of biliary acids essential for the digestion and absorption of fatty acids [243]. They also stimulate the secretion of digestive enzymes in the pancreas, including lipase, amylase, trypsin, and chymotrypsin, all of which play a crucial role in digestion [242]. For instance, fennel (*Foeniculum vulgare*) and ginger have long been used as traditional remedies for digestive health. In animal studies, fennel has been reported to stimulate the secretion of digestive enzymes, including amylase, lipase, and protease, from the pancreas, thus contributing to improved nutrient breakdown and absorption [244].

Being rich in phytonutrients and polyphenols, herbs and spices enhance microbial diversity and possess prebiotic-like properties, promoting the growth and activity of beneficial gut bacteria and thus fostering a resilient gut microbiota. A considerable body of evidence, mostly coming from animals, suggests that capsaicin exerts multiple benefits on the gut microbiota, involving various and complex mechanisms, targeting mainly metabolic and inflammatory diseases [245]. For example, in mice treated with capsaicin for one week, the relative abundances of Faecalibacterium were increased in the experimental group but not in the controls [246]. The gut bacterium *Faecalibacterium* (from *Bacillota phyla*), one of the most important symbiotic components of the human gut microbiome, is considered a bioindicator of human health, being negatively associated with IBD, immunity, obesity, diabetes, asthma, major depressive disorder, and colorectal cancer [247,248]. However, further confirmation of beneficial effects in humans is required. One small human trial (*n* = 12) that provided both low (5 mg/day) and high (10 mg/day) capsaicin concentrations administered for 2 weeks in healthy subjects led to an increase in the Bacillota/Bacteroidota ratio and *Faecalibacterium* abundance, accompanied with increased plasma levels of glucagon-like peptide 1 and gastric inhibitory polypeptide and decreased plasma ghrelin levels [249].

Curcumin has also been shown to interact with gut commensal microbes. In a non-alcoholic fatty liver disease rat model induced by a high-fat diet, rats were randomly divided into three groups: standard diet, high-fat diet, and high-fat diet plus curcumin [250]. Curcumin supplementation significantly shifted the composition of the microbiota toward that of the control rats fed a standard diet, counteracting the high-fat-diet-induced abundance of several genera that have previously been associated with diabetes and inflammation, such as *Ruminococcus* [250,251]. Other studies confirmed that curcumin shifted the balance in favour of beneficial bacteria strains, including *Bifidobacteria*, *Lactobacilli*, and butyrate-producing bacteria, while reducing pathogenic ones associated with systemic diseases, such as *Prevotellaceae*, *Coriobacterales, Enterobacteria*, and *Rikenellaceae* [251,252].

Ginger is also known for its microbe-modulating effects. In an obese mouse model induced by a high-fat diet, 6-gingerol, a bioactive compound of ginger, was shown to reshape the composition of the gut microbiota by increasing the abundance of weight-loss-associated genera (e.g., *Muribaculaceae*, *Alloprevotella*, and *Akkermansiaand*) and decreasing the abundance of obesity-contributed bacteria (e.g., *Lachnospiraceae* and *Lactobacillus reuteri*) [253]. Moreover, the effect of ginger-root powder was tested in healthy adults in a double-blind placebo-controlled trial, showing that while supplementation with ginger root powder induced changes in the gut microbiota composition, it did not change the microbial diversity, bowel function, gastrointestinal symptoms, mood, or quality of life [254].

In summary, the bioactive compounds in spices exhibit multifaceted effects on the gut, influencing the microbiota and promoting the production of beneficial bacteria and bacterial metabolites, possibly contributing to overall gut microbiota resilience.

### 6.6. Digestive Enzymes

It is worthwhile to briefly consider digestive enzymes in the context of functional food development for gastrointestinal health. The gastrointestinal system produces and releases digestive enzymes to break down fats, proteins, and carbohydrates, facilitating the digestion and subsequent absorption of nutrients. Supplementation with these enzymes, when deemed necessary, can serve as a dependable adjunct in the treatment of various digestive disorders [255]. Most digestive enzymes are available to consumers in the form of food supplements as pills or capsules. The inclusion of carefully selected digestive enzymes as additions to functional foods may provide a useful dietary adjunct to manage and support digestive health. Currently, a variety of enzyme supplementation formulations are accessible on the market and are actively employed in clinical practice to address numerous digestive diseases, such as exocrine pancreatic insufficiency in chronic pancreatitis, pancreatic cancer, cystic fibrosis (CF), diabetes, the management of lactose intolerance, celiac disease, and post-prandial distress [255,256]. Interestingly, the combination of probiotics and lactase supplementation was shown to reduce the symptoms of intolerance in patients with lactose malabsorption [257,258]. Therefore, tailored products containing enzyme supplements in addition to probiotics seem to offer an advantage to the therapeutic management of such disorders.

Finally, emerging evidence indicates that pancreatic exocrine insufficiency (PEI) may be associated with the gut microbiota, and patients with PEI display alterations in their gut microbiota [259]. In mice, oral supplementation with pancreatic enzymes was found to alter the gut microbiota, with both *A. muciniphila* and *L. reuteri* increasing their relative abundance [260]. The authors hypothesized that pancreatic enzymes may exert their effect via the gut microbiota, potentially increasing the abundance of strains which play a role in the maintenance of the intestinal barrier. Further research is required to determine how the provision of certain digestive enzymes might contribute to priming the gut for resilience via their effects on the gut microbiota.

## 7. Relevance to Future Research and Product Development

When designing foods with multiple ingredients, “casting a wide net” may prove beneficial, particularly given the high likelihood of inter-individual variability in responses to bioactive ingredients at the level of the gut. Such an approach, whereby product developers select a broad array of plant-based ingredients for gut health, may indeed contribute to improved microbiota diversity. Carefully designed functional foods, incorporating a variety of bioactive ingredients for gut health, may contribute to improved microbiota diversity, host physiology, and broader well-being. As we navigate the intricate landscape of nutrition, understanding these complex interactions provides a foundation for designing foods that may prime the gut for health and resilience. In the current landscape, the field stands at an exciting juncture with vast untapped potential. Although there is a wealth of studies utilising in vitro and in vivo models, as well as some promising but underpowered human trials showing potential health benefits, the need for future well-designed human intervention studies is paramount. These studies are essential for providing a more comprehensive and accurate understanding of the impact of functional food ingredients on human health. This ongoing exploration is crucial for contextualizing findings within the broader scope of the literature, advancing our understanding of how dietary choices can prime the gut for health and resilience.

## 8. Conclusions

In conclusion, this scientific review provides novel insights into the intricate interplay between bioactive food ingredients and human physiology, underscoring the potential for both local and systemic effects through their impact on gut health (see Figure 3). The adaptability of dietary compounds to exert specific influences locally in the gut, and broader impacts on the entire organism, emphasises the nuanced dynamics of nutritional choices. Crucially, our discussion emphasises the pivotal role of the gut microbiota in adapting to stress, with gut microbiota resilience serving as a cornerstone of the body’s adaptive capacity. Incorporating plant-derived functional foods, including dietary fibres, fermented plants, probiotics, and postbiotics, as well as polyphenols and bioactive compounds from spices, aligns with dietary hormesis—a concept where mild stressors promote adaptive responses, fostering gut priming for enhanced resilience and overall well-being. The potential for designing effective and targeted functional foods to support human health is immense, and future research endeavours will play a pivotal role in unlocking this potential by providing evidence-based insights that can inform dietary recommendations and enhance overall well-being.

## Figures and Tables

**Figure 1 foods-13-00739-f001:**
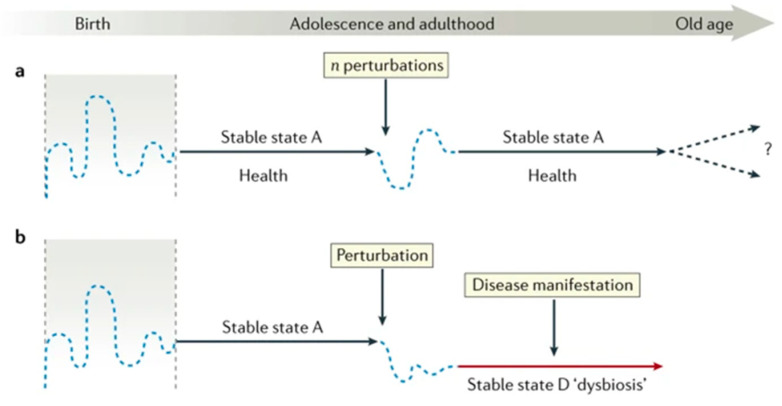
How perturbations (or challenges) may influence the homeostasis of the gut microbiota. (Figure reproduced with permission from Sommer et al., 2017 [16]).

**Figure 2 foods-13-00739-f002:**
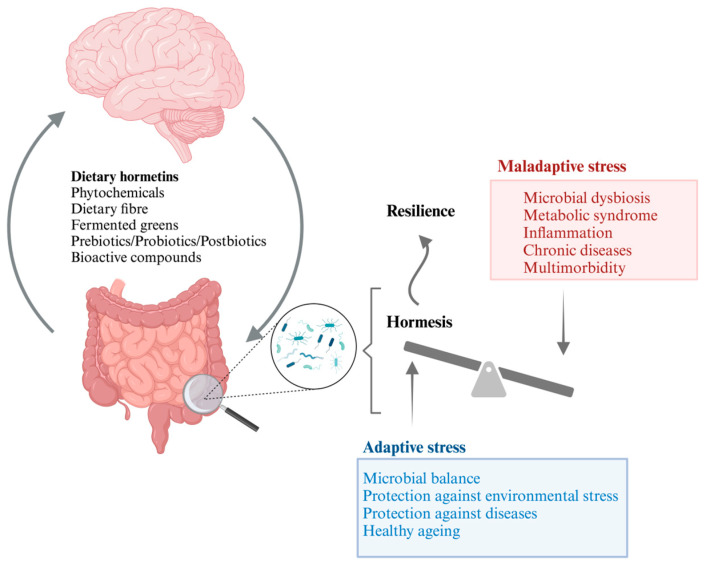
The relationship between dietary hormetins, gut microbiota resilience, and the bidirectional communication with the brain that may shape the body’s ability to respond to stress.

**Figure 3 foods-13-00739-f003:**
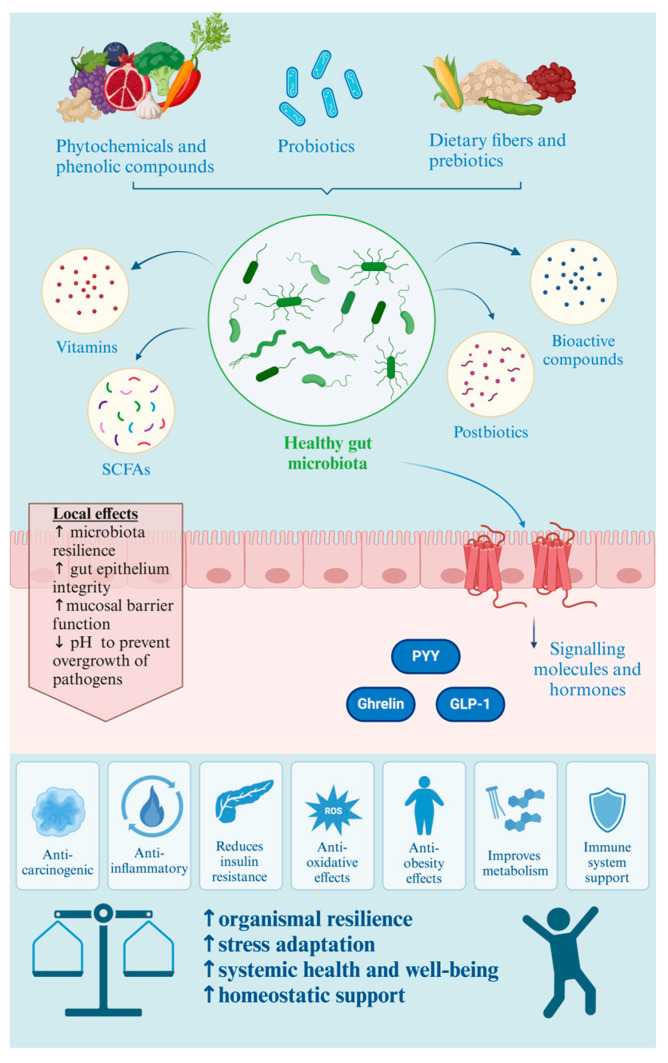
Functional food components can be carefully selected to exert positive effects on the gut microbiota, which may contribute to enhancing resilience and promoting homeostasis and stress adaptation.

## Data Availability

The original contributions presented in the study are included in the article material, further inquiries can be directed to the corresponding author.

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
