# Peer review of "Local and Systemic Effects of Bioactive Food Ingredients: Is There a Role for Functional Foods to Prime the Gut for Resilience?"

_foods, 2024, doi:10.3390/foods13050739_

Round 1

Reviewer 1 Report

Comments and Suggestions for Authors

I do not think the title of the manuscript includes the meaning of the text. The title is inappropriate.

Line 95, please delete the comma before reference 23

I suggest the authors provide higher quality figures. Currently, the resulotion of the figures is relatively low.

I suggest move figure 3 to section 7, not in conclusion section.

I am cofused about section 6. Dietary fibres, local and systemic effects of probiotics and post-biotics, plant based-fermented foods, poylphenols and phytonutrients, spices and aromatic herbs, and digestive enzymes. Are they parallel? Additionally, the content is too lengthy and boring.

Author Response

I do not think the title of the manuscript includes the meaning of the text. The title is inappropriate.

Thank you for expressing this opinion. Could you please clarify why you consider this to be inappropriate? This would help us to carefully consider this opinion from a scientific perspective and consider options for a new title.

Line 95, please delete the comma before reference 23
Done

I suggest the authors provide higher quality figures. Currently, the resulotion of the figures is relatively low.
We have inserted images with higher resolution

I suggest move figure 3 to section 7, not in conclusion section.
Done

I am cofused about section 6. Dietary fibres, local and systemic effects of probiotics and post-biotics, plant based-fermented foods, poylphenols and phytonutrients, spices and aromatic herbs, and digestive enzymes. Are they parallel? Additionally, the content is too lengthy and boring.
We have added an introductory sentence to constextualise this section. We are sorry you find the content too lengthy and boring. We have shortened the paper in certain places. You will be able to read the changes in the updated manuscript.

The content is aimed for food scientists that may not be familiar with the field and the myriad of potential ingredients available.

Reviewer 2 Report

Comments and Suggestions for Authors

This manuscript reviews the gut health effects of functional foods and their  ingredients,relevant research progress is discussed comprehensively, but the structure is somewhat confused. Not including the Introduction 1, the order seems to be more reasonable for part 4-2-3-5-6-7. The overall text of the manuscript is too much, and more paragraphs in the same part are not friendly to readers. It may be better to visualize and generalize the gut health function of functional foods.

Comments on the Quality of English Language

ok

Author Response

This manuscript reviews the gut health effects of functional foods and their  ingredients,relevant research progress is discussed comprehensively, but the structure is somewhat confused. Not including the Introduction 1, the order seems to be more reasonable for part 4-2-3-5-6-7.

  • Thank you for this helpful suggestion, we have modified the order accordingly.

The overall text of the manuscript is too much, and more paragraphs in the same part are not friendly to readers. It may be better to visualize and generalize the gut health function of functional foods.

  • We have visualised and generalised the gut health function of bioactive ingredients in functional foods in Figure 3. We have removed the section

6.5.2 Potential systemic health benefits of herbs and spices against chronic diseases.

  • We have shortened certain sections which you will find in the updated version attached in the system.

Reviewer 3 Report

Comments and Suggestions for Authors

The paper is a review about the recent studies in the field of bioactive food ingredients that are also functional foods and prime the gut for resilience.

The review highlights new perspectives on the complex interactions between bioactive food ingredients and the human body showing their local effects as well as systemic effects through their impact on intestinal health.

The review is well documented and structured presenting more interesting subjects related to the impact of the gut microbiome on host health, the interaction between diet, microbiota and human health giving in details information on the functional food and beverage ingredients with promising effects on the gut health.

Observations:

1.     The method of reviewing the literature was not shown. I recommend to present details on the key words used to find the information reviewed in the paper.

2.     The reference for the figures 2 and 3 should be added.

3.     The subdivisions of subchapter “6.5.2 Potential systemic health benefits of herbs and spices against chronic diseases should be corrected”.  

Author Response

The paper is a review about the recent studies in the field of bioactive food ingredients that are also functional foods and prime the gut for resilience.

The review highlights new perspectives on the complex interactions between bioactive food ingredients and the human body showing their local effects as well as systemic effects through their impact on intestinal health.

The review is well documented and structured presenting more interesting subjects related to the impact of the gut microbiome on host health, the interaction between diet, microbiota and human health giving in details information on the functional food and beverage ingredients with promising effects on the gut health.

Observations:

  1. The method of reviewing the literature was not shown. I recommend to present details on the key words used to find the information reviewed in the paper.
  • This is not a formal requirement for a review paper in the Foods journal, therefore we have not included a literature review and methodology section. Such a section would also extend the length of the paper and several other reviewers asked to shorten the paper.

      2. The reference for the figures 2 and 3 should be added.

  • They are novel figures we have created ourselves.

      3. The subdivisions of subchapter “6.5.2 Potential systemic health benefits of herbs and spices against chronic diseases should be corrected”. 

  • This section has been deleted following comments from another reviewer

Reviewer 4 Report

Comments and Suggestions for Authors

Manuscript ID: foods- 2846528

Title: Local and systemic effects of bioactive food ingredients: Is there a role for functional foods to prime the gut for resilience?

Dear Editor

The manuscript is about the gut microbiota and functional ingredient that can improve their resilience. The manuscript gives useful information and well written. However, it needs major revision to improve its quality according bellow:

·       Line 154: SCFAs?? Please use abbreviations after first appearance.

·       Some parts need discussion in depth. For example, Fig 1, Line 246-259. Line 273-279.

·       Moreover, I suggest authors mention the role of gut microbiota on lung function and prevention of some lung disorders such as COVID_19.

·       Figure captain should be short and the discussion should be in the manuscript.

·       Dietary fibers:  please insert source of polysaccharides as prebiotics, such as algal based polysaccharides like fucoidan.

·       Line 641: Saccharina japonica, specific names should be italic

·       Line 684: “Polyphenols and phytonutrients “section: what about the effect of phlorotannins on gut microbiota?

·       What about the role of carotenoids from various sources such as fucoxanthin, astaxanthin, etc on gut microbiota?

·       There some probiotic foods. Please insert some information about their popularity, and market? In which food industries probiotic are more successful?

·       Line 874-908: “6.5.2 Potential systemic health benefits of herbs and spices against chronic diseases “section. These parts reveal useful information but not related to the main subject of the manuscript.

·       Conclusion: please insert Fig 3 in right place.  

·       English should be improved.

Comments on the Quality of English Language

Manuscript ID: foods- 2846528

Title: Local and systemic effects of bioactive food ingredients: Is there a role for functional foods to prime the gut for resilience?

Dear Editor

The manuscript is about the gut microbiota and functional ingredient that can improve their resilience. The manuscript gives useful information and well written. However, it needs major revision to improve its quality according bellow:

·       Line 154: SCFAs?? Please use abbreviations after first appearance.

·       Some parts need discussion in depth. For example, Fig 1, Line 246-259. Line 273-279.

·       Moreover, I suggest authors mention the role of gut microbiota on lung function and prevention of some lung disorders such as COVID_19.

·       Figure captain should be short and the discussion should be in the manuscript.

·       Dietary fibers:  please insert source of polysaccharides as prebiotics, such as algal based polysaccharides like fucoidan.

·       Line 641: Saccharina japonica, specific names should be italic

·       Line 684: “Polyphenols and phytonutrients “section: what about the effect of phlorotannins on gut microbiota?

·       What about the role of carotenoids from various sources such as fucoxanthin, astaxanthin, etc on gut microbiota?

·       There some probiotic foods. Please insert some information about their popularity, and market? In which food industries probiotic are more successful?

·       Line 874-908: “6.5.2 Potential systemic health benefits of herbs and spices against chronic diseases “section. These parts reveal useful information but not related to the main subject of the manuscript.

·       Conclusion: please insert Fig 3 in right place.  

·       English should be improved.

Author Response

Dear Editor

The manuscript is about the gut microbiota and functional ingredient that can improve their resilience. The manuscript gives useful information and well written. However, it needs major revision to improve its quality according bellow:

  •  Line 154: SCFAs?? Please use abbreviations after first appearance.

Short-chain fatty acids are now abbreviated following their first mention.

  •  Some parts need discussion in depth. For example, Fig 1, Line 246-259. Line 273-279.

Please find this updated section appearing earlier in the manuscript according to reviewers recommendations where more discussion has been added as per your suggestion.

  •  Moreover, I suggest authors mention the role of gut microbiota on lung function and prevention of some lung disorders such as COVID_19.

Since we are trying to shorten the paper, in light of yours and other reviewers comments, we are concerned that this angle might detract the reader from the theme of the paper. Since we haven’t gone into depth on other gut-organ axis we also consider this to be too far from the thread of the paper.

  •  Figure captain should be short and the discussion should be in the manuscript.

Figure captions have been shortened and the extra detail has been explained in the text.

  •  Dietary fibers:  please insert source of polysaccharides as prebiotics, such as algal based polysaccharides like fucoidan.

This has been added.

  •  Line 641: Saccharina japonica, specific names should be italic
    Done
  •  Line 684: “Polyphenols and phytonutrients “section: what about the effect of phlorotannins on gut microbiota?

This has been added.

  •  What about the role of carotenoids from various sources such as fucoxanthin, astaxanthin, etc on gut microbiota?

This has been added.

  •  There some probiotic foods. Please insert some information about their popularity, and market? In which food industries probiotic are more successful?

This has been added at line 488.

  •  Line 874-908: “6.5.2 Potential systemic health benefits of herbs and spices against chronic diseases “section. These parts reveal useful information but not related to the main subject of the manuscript.

We have removed this section

  •  Conclusion: please insert Fig 3 in right place.  

Done

  •  English should be improved.

We have conducted a review with 2 native English speakers and will use the English language service provided by the journal if deemed necessary.